# AN AUTOREGRESSIVE FLOW MODEL FOR 3D MOLECULAR GEOMETRY GENERATION FROM SCRATCH

**Youzhi Luo & Shuiwang Ji**
Department of Computer Science & Engineering
Texas A&M University
College Station, TX 77843, USA
`{yzluo, sji}@tamu.edu`

## ABSTRACT

We consider the problem of generating 3D molecular geometries from scratch. While multiple methods have been developed for generating molecular graphs, generating 3D molecular geometries from scratch is largely under-explored. In this work, we propose G-SphereNet, a novel autoregressive flow model for generating 3D molecular geometries. G-SphereNet employs a flexible sequential generation scheme by placing atoms in 3D space step-by-step. Instead of generating 3D coordinates directly, we propose to determine 3D positions of atoms by generating distances, angles and torsion angles, thereby ensuring both invariance and equivariance. In addition, we propose to use spherical message passing and attention mechanism for conditional information extraction. Experimental results show that G-SphereNet outperforms previous methods on random molecular geometry generation and targeted molecule discovery tasks. Our code is publicly available as part of the DIG package (`https://github.com/divelab/DIG`).

## 1 INTRODUCTION

Designing and synthesizing novel molecules with desirable properties is a challenging task in drug discovery (Wang et al., 2022; Stokes et al., 2020) and chemical science. The size of search space of all chemical molecules is estimated to be on the order of $10^{33}$ (Polishchuk et al., 2013), thereby making exhaustive search infeasible. In recent years, advances in machine learning methods have greatly accelerated the progress in this filed. Many studies represent molecules as 2D molecular graphs and propose to automatically generate molecular graphs and optimize molecular properties with deep generative models, such as variational auto-encoders (Kingma & Welling, 2014).

However, complete information of molecules cannot be obtained from 2D molecular graphs because 3D structures, also known as 3D molecular geometries, are critical in determining many molecular properties. 3D molecular geometries represent the 3D coordinates of atoms and are important for the accurate prediction of quantum properties (Schütt et al., 2017). Hence, we argue that generating 2D molecular graphs might not be the best way to identify novel molecules with certain desirable quantum properties. Instead, developing a generative model that can generate 3D molecular geometries from scratch is a promising solution to this problem. Currently, this area remains under-explored. Recently, a series of seminal studies (Xu et al., 2021b;a; Shi et al., 2021) have proposed to generate 3D molecular geometries from given 2D molecular graphs. These methods themselves do not generate novel molecules once the 2D molecular graphs are given.

In this work, we propose G-SphereNet, a Generative model for 3D molecular geometry generation from scratch inspired by SphereNet (Liu et al., 2022). In G-SphereNet, 3D molecular geometries are generated by sequentially placing atoms in 3D space. The 3D positions of atoms are implicitly determined by generating distances, angles and torsion angles to ensure both invariance and equivariance properties. Our work is inspired by SphereNet, which use distances, angles and torsion angles to compute predictive representations of molecules. In addition, G-SphereNet employs SphereNet and attention mechanism to extract conditional information. Experimental results show that G-SphereNet can outperform prior methods on the 3D molecular geometry generation task.

## 2 BACKGROUND AND RELATED WORK

### 2.1 2D MOLECULAR GRAPH GENERATION

Recently, thanks to the advances of deep generative models, significant progress has been made in the problem of molecule design and generation. Some methods (Gómez-Bombarelli et al., 2018; Kusner et al., 2017; Dai et al., 2018; Liu et al., 2020) use sequence models to generate SMILES (Weininger, 1988) string representations of molecules. Other studies consider molecules as graphs, in which atoms and chemical bonds of the molecule are represented by nodes and edges, respectively. These studies either generate the node type and adjacency matrix of the graph (Simonovsky & Komodakis, 2018; Ma et al., 2018; De Cao & Kipf, 2018; Liu et al., 2021c), or form the molecular graph by sequentially adding nodes and edges (You et al., 2018; Shi* et al., 2020), or compose the molecule from a junction tree of molecular motifs (Jin et al., 2018).

However, these methods only generate the graph structures of molecules while crucial 3D molecular geometries are ignored. In other words, the 3D coordinates of atoms in the molecule are unknown. Hence, these generative models cannot distinguish spatial isomers, i.e., molecules with the same molecular graph but different 3D molecular geometries. In addition, 3D molecular geometries are required for the computation of some quantum properties of molecules, such as HOMO-LUMO gap (Hu et al., 2021; Liu et al., 2021a; Ying et al., 2021; Addanki et al., 2021; Xu et al., 2021c). Hence, these generation methods cannot be used when spatial isomers or quantum properties are needed.

### 2.2 3D MOLECULAR GEOMETRY GENERATION FROM 2D INFORMATION

Currently, some studies have proposed to generate 3D molecular geometries from the 2D conditional information of target molecules. Some methods (Simm et al., 2020; 2021) propose to generate 3D molecular geometries through minimizing the energy of atomic systems, in which the numbers and types of atoms are explicitly given in atom bags. Other methods (Mansimov et al., 2019; Simm & Hernandez-Lobato, 2020; Gogineni et al., 2020; Xu et al., 2021b;a; Shi et al., 2021; Ganea et al., 2021) propose to randomly sample multiple 3D molecular geometries from the molecular graph of the target molecule with deep generative models. These conditional generation methods assume that we are given the target molecules in the form of molecular graphs, but we do not know their 3D geometries. However, this assumption does not hold in the targeted molecule discovery problem. In this problem, we aim to discover novel molecules with good quantum properties, such as low HOMO-LUMO gaps. In other words, target molecules themselves are unknown and to be generated, and their 3D geometries are also needed because quantum properties cannot be accurately estimated only from molecular graphs. Hence, we argue that developing a method to generate 3D molecular geometries from scratch is more useful to this problem.

### 2.3 3D MOLECULAR GEOMETRY GENERATION FROM SCRATCH

In this work, we consider the problem of generating 3D molecular geometries from scratch. Let $\mathcal{G} = \{G_j\}_{j=1}^m$ be a set of 3D molecular geometries, and the function $S(G) \in \mathbb{R}$ computes a specific quantum property score of $G$. We consider the two generation tasks defined as below.

- Learning a random generation model $p_\theta(\cdot)$ from $\mathcal{G}$, so that the model can sample a valid 3D molecular geometry $G$ with a high probability $p_\theta(G)$.

- Learning a targeted molecule discovery model $p_\theta(\cdot)$ so as to maximize (or minimize) the expected quantum property score $\mathbb{E}_{G \sim p_\theta}[S(G)]$.

This problem is largely under explored and there are only a few studies attempting to solve it. G-SchNet (Gebauer et al., 2019) uses an autoregressive model based on SchNet (Schütt et al., 2017) to sequentially generate the new atom and place it at the local grid point of a focal atom. On the other hand, EDMNet (Hoffmann & Noé, 2019) and 3DMolNet (Nesterov et al., 2020) generate pairwise distances between atoms with generative adversarial networks (GAN) (Goodfellow et al., 2014) and variational auto-encoders (VAE) (Kingma & Welling, 2014), respectively. Besides, E-NFs (Satorras et al., 2021a) proposes a geometry generation model by combining flow models (Rezende & Mohamed, 2015) with E(n) equivariant graph neural networks (Satorras et al., 2021b). It generates 3D coordinates of all atoms in a one-shot fashion and defines the prior distribution in a subspace of

the latent space to ensure translation invariance. Different from these methods, our method employs a flexible sequential generation pipeline based on autoregressive flow models, which can capture the density of 3D molecular geometries more effectively.

## 2.4 FLOW MODELS

A flow model defines a parametric invertible mapping $f_\theta : z \in \mathbb{R}^d \to x \in \mathbb{R}^d$, where the data point $x$ and the latent variable $z$ are both random variables. Given that $z$ is sampled from a known prior distribution $p_Z$ and $f_\theta$ is invertible, we can compute the log-likelihood of $x$ as

$$\log p_X(x) = \log p_Z\left(f_\theta^{-1}(x)\right) + \log\left|\det J\right|, \tag{1}$$

where $J = \frac{\partial f_\theta^{-1}(x)}{\partial x}$ is the Jacobian matrix. To train the flow model on a given dataset $\mathcal{X} = \{x_i\}_{i=1}^m$, the log-likelihoods of data points are computed from Eqn. (1) and maximized by gradient descent. Hence, tractable and cheap computation of $\det J$ is needed for efficient training. A common choice of $f_\theta$ in most flow models is the affine coupling mapping (Dinh et al., 2014; 2016), in which case computing $\det J$ is very easy because $J$ is an upper triangular matrix.

Flow models have been used in a variety of generation tasks (Rezende & Mohamed, 2015; Tran et al., 2019; Köhler et al., 2020). Compared with VAE and GAN, they allow for exact likelihood computation, and can model the density of data more accurately. Because of these advantages, many recent studies have used flow models in the molecule generation task. Some one-shot methods, including GraphNVP (Madhawa et al., 2019), GRF (Honda et al., 2019), and MoFlow (Zang & Wang, 2020), consider the node type and adjacency matrix as the generation targets. On the other hand, GraphAF (Shi* et al., 2020) and GraphDF (Luo et al., 2021) generate molecular graphs by generating nodes and edges sequentially through autoregressive flow models (Papamakarios et al., 2017). These models have stronger capacity to model graph structures than one-shot methods, and achieve state-of-the-art performance in the molecular graph generation task.

## 3 METHODS

While autoregressive flow models have been successfully applied to the molecular graph generation task, it remains unclear whether they are sufficiently powerful to model more complicated 3D molecular geometries. In this section, we present G-SphereNet, a novel 3D molecular geometry generation method. It adopts a flexible, effective, and efficient sequential generation pipeline based on autoregressive flow models, which can ensure the equivariance property of coordinates and the invariance property of likelihood simultaneously. In addition, expressive spherical message passing based graph neural networks (Liu et al., 2022) and multi-head attention networks (Vaswani et al., 2017) are used in G-SphereNet to extract 3D conditional information for accurate generation. To the best of our knowledge, G-SphereNet is the first likelihood-based autoregressive generative model for 3D molecular geometry generation.

## 3.1 SEQUENTIAL GENERATION

Let $k$ be the number of atom types. We use a 3D point cloud $G = (A, R)$ to represent the 3D geometry of a molecule with $n$ atoms, where $A \in \{0, 1\}^{n \times k}$ is the atom type matrix and $R \in \mathbb{R}^{n \times 3}$ is the atom coordinate matrix. Each row in the matrix $A$ is a one-hot vector, and $A[j, u] = 1$ represents that the $j$-th atom has type $u$. The 3-dimensional row vector at the $j$-th row of the matrix $R$ represents the 3D Cartesian coordinate of the $j$-th atom.

We consider the generation of 3D molecular geometries as a sequential decision process. We start from a molecular geometry $G_1$ with one carbon atom at the origin point, and generate the complete geometry by adding a new atom at each step. Specifically, at the $i$-th step, let the intermediate 3D molecular geometry generated from the previous $i-1$ steps be $G_i = (A_i, R_i)$, which has $i$ atoms. The atom type $a_i \in \{0, 1\}^k$ of the new atom is generated by the generative model $g^a$ based on the latent variable $z_i^a$. Afterwards, the generative model $g^r$ decides the 3D Cartesian coordinate $r_i \in \mathbb{R}^3$ of the new atom based on the latent variable $z_i^r$. Both $g^a$ and $g^r$ are autoregressive functions of intermediately generated geometries. The overall sequential generation process can be described by the following equations:

$$a_i = g^a\left(z_i^a; A_i, R_i\right), \; r_i = g^r\left(z_i^r; A_i, R_i\right), \quad i \geq 1. \tag{2}$$

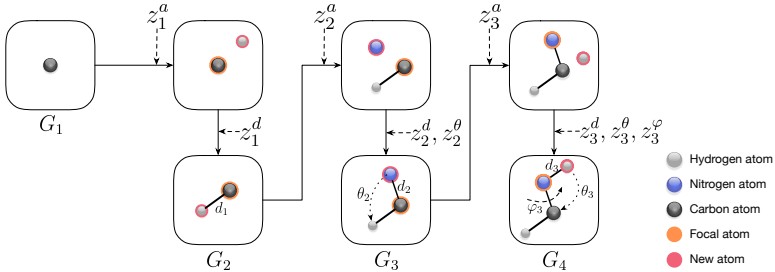

Figure 1: An illustration of the sequential generation process in G-SphereNet.

## 3.2 GENERATION WITH AUTOREGRESSIVE FLOW MODELS

We employ autoregressive flow models to generate the atom type $a_i$ of the new atom at each step. Since atom types are discrete numbers, we adopt the dequantization method (You et al., 2018; Shi* et al., 2020) to convert them into continuous numbers by adding real-valued noise as

$$\tilde{a}_i = a_i + u, u \sim U(0,1)^k, \quad i \geq 1, \tag{3}$$

where $U(0,1)$ is the uniform distribution over the interval $(0,1)$. To generate $a_i$, we first sample the latent variable $z_i^a \in \mathbb{R}^k$ from the standard Gaussian distribution $\mathcal{N}(0,1)$, and then map $z_i^a$ to $\tilde{a}_i$ by the affine transformation as

$$\tilde{a}_i = s_i^a \odot z_i^a + t_i^a, \quad i \geq 1, \tag{4}$$

where $\odot$ denotes the element-wise multiplication, the scale factor $s_i^a$ and the shift factor $t_i^a$ both depend on the conditional information extracted from the intermediate geometry $G_i = (A_i, R_i)$. Intuitively, $\tilde{a}_i$ should be invariant to any rigid transformation on $R_i$, i.e., $\tilde{a}_i$ should not change if we rotate or translate $R_i$ in 3D space. Hence, we use a symmetry-invariant model to compute $s_i^a$ and $t_i^a$ from $G_i$, which are described in Sec. 3.4 in detail. After obtaining $\tilde{a}_i$, $a_i$ can be computed by taking the argmax of $\tilde{a}_i$, as $a_i = \text{one-hot}(\arg\max \tilde{a}_i)$.

However, we cannot generate the 3D coordinate $r_i$ in the same way as the generation of the atom type $a_i$. Directly calculating $r_i$ with the autoregressive flow model, as in Eqn. (4), neither satisfies the equivariance property of coordinates nor the invariance property of likelihood. First, it is easy to find that if we rotate or translate $R_i$, then $r_i$ needs to be rotated or translated correspondingly. Formally, it means that for any orthogonal matrix $Q \in \mathbb{R}^{3 \times 3}$ and translation vector $b \in \mathbb{R}^3$, if $r_i = g^r(z_i^r; A_i, R_i)$, we have

$$Qr_i + b = g^r \left( z_i^r; A_i, R_i Q^T + \mathbf{1} b^T \right), \tag{5}$$

where $\mathbf{1}$ denotes a vector of all ones of length $i$. If we directly compute $r_i$ by the autoregressive model similar to Eqn. (4), i.e., $r_i = s_i^r \odot z_i^r + t_i^r$, then to satisfy Eqn. (5), the correctness of $s_i^{Qr+b} \odot z_i^r + t_i^{Qr+b} = Q[s_i^r \odot z_i^r] + Qt_i^r + b$ has to be ensured for any orthogonal matrix $Q$ and translation vector $b$. However, it is very hard to design a flow model satisfying this complicated condition. Second, the likelihood $p(r_i | A_i, R_i)$ should be invariant to rotations and translations as they do not change the 3D structure. In other words,

$$p \left( Qr_i + b | A_i, R_i Q^T + \mathbf{1} b^T \right) = p \left( r_i | A_i, R_i \right) \tag{6}$$

should be satisfied for any orthogonal matrix $Q$ and translation vector $b$. It follows from the change-of-variable theorem that $p \left( r_i | A_i, R_i \right) = p \left( z_i^r \right) \left| \det \frac{\partial z_i^r}{\partial r_i} \right|$. Let the latent variable corresponding to $Qr_i + b$ be $z_i^{Qr+b}$. The affine transformation in Eqn. (4) does not admit any relationship between either $p(z_i^r)$ and $p(z_i^{Qr+b})$, or $\frac{\partial z_i^r}{\partial r_i}$ and $\frac{\partial z_i^{Qr+b}}{\partial (Qr_i+b)}$, so Eqn. (6) is not guaranteed to hold. Hence, we cannot guarantee the invariance property.

Given these limitations, we instead propose to determine the 3D relative position of the new atom by symmetry-invariant elements. Similar to G-SchNet (Gebauer et al., 2019), we first choose a focal atom among all $i$ atoms in $G_i$, which serves as the reference point for the location of the new atom. Then, the model generates the distance $d_i$, the angle $\theta_i$, and the torsion angle $\varphi_i$ successively. Specifically, assuming that the focal atom is the $f$-th atom of $G_i$, the distance $d_i = ||r_i - r_f||_2$ from the focal atom to the new atom is first generated. Then, if $i \geq 2$, the angle $\theta_i \in [0, \pi]$ between the

lines $(r_f, r_i)$ and $(r_f, r_c)$ is generated, where $c$ is the atom closest to $f$ in $G_i$. Finally, if $i \geq 3$, the torsion angle $\varphi_i \in [-\pi, \pi]$ between the planes formed by positions $(r_f, r_c, r_i)$ and $(r_f, r_c, r_e)$ is generated, where $e$ is the atom closest to $c$ but different from $f$ in $G_i$. Similar to $\tilde{a}_i$, $d_i$, $\theta_i$, and $\varphi_i$ are generated as

$$
\begin{aligned}
d_i &= s_i^d \odot z_i^d + t_i^d, \quad i \geq 1, \\
\theta_i &= s_i^\theta \odot z_i^\theta + t_i^\theta, \quad i \geq 2, \\
\varphi_i &= s_i^\varphi \odot z_i^\varphi + t_i^\varphi, \quad i \geq 3,
\end{aligned}
\tag{7}
$$

where $z_i^d, z_i^\theta, z_i^\varphi \in \mathbb{R}$ are all latent variables sampled from standard Gaussian distributions, and the scale factors $s_i^d, s_i^\theta, s_i^\varphi \in \mathbb{R}$ and the shift factors $t_i^d, t_i^\theta, t_i^\varphi \in \mathbb{R}$ are functions of $G_i$. Afterwards, the coordinate $r_i$ is computed from the relative positional elements $d_i, \theta_i, \varphi_i$ and the coordinates $r_f, r_c, r_e$. We show that such a process of placing the new atom in 3D space can strictly satisfy the conditions in Eqn. (5) and (6) in Appendix A, thereby satisfying both invariance and equivariance properties. The sequential generation process repeats until either the maximum number of atoms is reached, or no atom can be chosen as the focal atom by the atom-wise classifier. Our proposed sequential generation method is related to SphereNet (Liu et al., 2022) in that our generation targets are the 3D information used by SphereNet to extract features, so we name our method as G-sphereNet. An illustration of the overall generation process in G-SphereNet is given in Figure 1.

## 3.3 DISCUSSIONS

We argue that our proposed G-SphereNet method has many advantages over previous 3D molecular geometry generation methods. First, it is easier for G-SphereNet to generate valid geometries theoretically because the exact 3D coordinate of each atom can always be obtained. However, EDMNet (Hoffmann & Noé, 2019) and 3DMolNet (Nesterov et al., 2020) generate the pairwise distances of atoms in the form of distance matrices. There is no theoretical guarantee that the generated matrices are always valid Euclidean distance matrices, or correspond to coordinates in 3D space (Dokmanic et al., 2015). Second, the generation of 3D positions in G-SphereNet is more flexible than that in G-SchNet (Gebauer et al., 2019). In G-SchNet, the new atom has to be placed at one of the candidate grid points circling around the focal atom, but it can be placed at any relative position of the focal atom in G-SphereNet. Third, compared with E-NFs (Satorras et al., 2021a), G-SphereNet is more efficient and effective. E-NFs maps latent variables to 3D coordinates of atoms by a flow model. Since coordinates are not translation invariant, E-NFs proposes to obtain latent variables by computationally expensive operations. Specifically, E-NFs first samples from the prior distribution defined in a subspace of the latent space, then maps the sampled variable to the latent variable by a linear projection. In contrast, G-SphereNet obtain 3D positions by generating distances, angles and torsion angles, which are naturally translation invariant. Hence, G-SphereNet can avoid the complicated operations of E-NFs. In addition, E-NFs generates the coordinates of all atoms in the geometry at a time, while G-SphereNet obtains the coordinate of one atom at a time. Though G-SphereNet may be slower, we argue that the sequential generation fashion helps the model to capture the dependency between atoms and the density of geometries more effectively. Experimental results also demonstrate that G-SphereNet can generate much more valid molecular geometries than E-NFs.

## 3.4 CONDITIONAL INFORMATION EXTRACTION

As we have mentioned in Sec. 3.2, generating the atom type and 3D position of the new atom requires capturing conditional information from the intermediate geometry at each step. Desirable conditional information should incorporate comprehensive 3D structural features of the geometry, and be invariant to any rigid transformation. To achieve this goal, we propose to capture conditional information using SphereNet (Liu et al., 2022), an advanced 3D graph neural network model. SphereNet considers the input molecular geometry as a directional cutoff graph. Denoting the feature of $k$-th edge by $e_k$ and the feature of $i$-th node by $v_i$, SphereNet initializes them with spherical basis functions and updates them to $e_k'$ and $v_i'$ with spherical message passing as

$$
\begin{aligned}
e_k' &= \phi^e \left( e_k, v_{r_k}, v_{s_k}, E_{s_k}, \rho^{p \to e} \left( R_{s_k} \right) \right), \\
v_i' &= \phi^v \left( v_i, \rho^{e \to v} (E_i) \right).
\end{aligned}
\tag{8}
$$

Here, $\phi^e, \phi^v$ are updating functions, $\rho^{p \to e}, \rho^{e \to v}$ are aggregation functions, $r_k$ and $s_k$ are the sending and receiving nodes of the $k$-th edge, $R_{s_k}$ are the coordinates of neighboring nodes of $s_k$, $E_{s_k}$ and

$\boldsymbol{E_i}$ are the features of edges incident to node $s_k$ and $i$, separately. SphereNet has powerful 3D structural feature extraction ability, and achieves good performance in multiple quantum property prediction tasks. Considering these advantages, we use the SphereNet model as the backbone feature extractor to capture conditional information from the intermediate molecular geometry.

For the input molecular geometry $G_i$, let the node embeddings computed from the SphereNet be $\{h_{i,j}\}_{j=0}^{i-1}$. To select the focal atom, we use an atom-wise multi-layer perceptron (MLP) taking the corresponding node embedding as input, and randomly choose the focal atom $f$ from atoms whose classification scores are higher than 0.5. If all the classification scores outputted from the atom-wise classifier are lower than 0.5, then no atom can be chosen as the focal atom and the sequential generation process terminates. Afterwards, the scale and shift factors in Eqn. (4) and (7) are computed. We ever tried computing them using solely the node embeddings of the reference points $f$, $c$, and $e$. However, our experiments show that it frequently causes incorrect placements of new atoms in 3D space. We think it is because node embeddings only contain local 3D information, which is insufficient for the accurate generation of the 3D positions of new atoms.

To tackle the above issue, we propose to augment node embeddings with global features extracted by a multi-head attention network (Vaswani et al., 2017). Formally, a multi-head attention network takes the query matrix $Q$, key matrix $K$, and value matrix $V$ as inputs, and extract global information from inputs by the multi-head attention mechanism:

$$Q_i = QW_i^Q, \ K_i = KW_i^K, \ V_i = VW_i^V, \ \text{ATT}_i = \text{softmax}\left(\frac{Q_iK_i^T}{\sqrt{p}}\right)V_i, \quad 1 \leq i \leq o,$$
$$\text{MH-ATT}(Q, K, V) = \text{Con}\left(\text{ATT}_1, ..., \text{ATT}_o\right)W^O, \tag{9}$$

where $\text{Con}(\cdot)$ denotes the concatenation operation, $p$ is the size of the second dimension of $K$, $o$ is the number of attention heads, the matrices $W_i^Q$, $W_i^K$, $W_i^V$, and $W^O$ are all trainable parameters. Let the node embedding matrix of $G_i$ be $H_i = [h_{i,0}, ..., h_{i,i-1}]^T$, $s_i^a$ and $t_i^a$ are computed as

$$\text{Con}(s_i^a, t_i^a) = \text{MLP}^a\left(\text{Con}\left(h_{i,f}, \text{MH-ATT}^a\left(h_{i,f}^T, H_i, H_i\right)\right)\right), \quad i \geq 1, \tag{10}$$

where $\text{MH-ATT}^a$ is a multi-head attention network and $\text{MLP}^a$ is a multi-layer perceptron. As for the scale and shift factors in Eqn. (7), we first multiply node embeddings with the atom type embedding vector to include the atom type information:

$$h_i^a = \text{Embedding}(a_i), \ \widetilde{h}_{i,j} = h_i^a \odot h_{i,j}, \ \widetilde{H}_i = \left[\widetilde{h}_{i,0}, ..., \widetilde{h}_{i,i-1}\right]^T, \quad i \geq 1, \ 0 \leq j \leq i - 1,$$

then compute them as

$$\text{Con}(s_i^d, t_i^d) = \text{MLP}^d\left(\text{Con}\left(\widetilde{h}_{i,f}, \text{MH-ATT}^d\left(\widetilde{h}_{i,f}^T, \widetilde{H}_i, \widetilde{H}_i\right)\right)\right), \quad i \geq 1,$$
$$\text{Con}(s_i^\theta, t_i^\theta) = \text{MLP}^\theta\left(\text{Con}\left(\widetilde{h}_{i,f}, \widetilde{h}_{i,c}, \text{MH-ATT}^\theta\left(\widetilde{h}_{i,f}^T, \widetilde{H}_i, \widetilde{H}_i\right)\right)\right), \quad i \geq 2, \tag{11}$$
$$\text{Con}(s_i^\varphi, t_i^\varphi) = \text{MLP}^\varphi\left(\text{Con}\left(\widetilde{h}_{i,f}, \widetilde{h}_{i,c}, \widetilde{h}_{i,e}, \text{MH-ATT}^\varphi\left(\widetilde{h}_{i,f}^T, \widetilde{H}_i, \widetilde{H}_i\right)\right)\right), \quad i \geq 3.$$

Here $\text{Embedding}(\cdot)$ is a lookup based embedding layer, $\text{MH-ATT}^d, \text{MH-ATT}^\theta, \text{MH-ATT}^\varphi$ are all multi-head attention networks, and $\text{MLP}^d, \text{MLP}^\theta, \text{MLP}^\varphi$ are all multi-layer perceptrons. The use of the multi-head attention networks helps extract more comprehensive 3D conditional information and is demonstrated to improve the generation performance a lot in our ablation study.

## 3.5 TRAINING

To train the G-SphereNet model on a dataset, we first need to split each individual molecular geometry in the dataset into a trajectory of atom addition steps. In other words, the generation order of atoms in the geometry and all corresponding focal atoms need to be determined. Since the generated atom is supposed to be placed in the local region of the focal atom during generation, we propose to obtain the training trajectory by applying Prim's algorithm on the geometry. This procedure can ensure that the sampled focal atom is always the nearest neighbor of the new atom among all atoms in the intermediate geometry.

Table 1: Comparison of different methods on the random molecular geometry generation task. The performance is evaluated by the chemical validity percentage and MMD distances of bond length distributions. Here ↑ means higher value indicates better performance, while ↓ means the opposite.

| Method | Validity↑ | MMD distances↓ | | | | | | |
|--------|-----------|-----|-----|-----|-----|-----|-----|---------|
| | | C-C | C-N | C-O | H-C | H-N | H-O | Average |
| E-NFs | 39.77% | 0.775 | 0.209 | 1.218 | 1.218 | 1.748 | 2.478 | 1.274 |
| G-SchNet | 81.49% | **0.183** | **0.078** | 0.320 | 1.236 | 1.396 | 2.399 | 0.935 |
| G-SphereNet | **88.18%** | 1.144 | 0.315 | **0.305** | **0.139** | **1.029** | **0.831** | **0.627** |

For a 3D molecular geometry $G$ having $n$ atoms $(n > 3)$, we maximize its log-likelihood to train the G-SphereNet model. Specifically, we obtain the generation targets, i.e., the atom type and 3D position of the atom to be generated at each step, then compute the log-likelihood of $G$ as

$$
\log p(G) = \sum_{i=1}^{n-1} \left[ \log p_{Z_a}(z_i^a) + \log \left| \frac{\partial \tilde{a}_i}{\partial z_i^a} \right| \right] + \sum_{i=1}^{n-1} \left[ \log p_{Z_d}(z_i^d) + \log \left| \frac{\partial d_i}{\partial z_i^d} \right| \right]
$$
$$
+ \sum_{i=2}^{n-1} \left[ \log p_{Z_\theta}(z_i^\theta) + \log \left| \frac{\partial \theta_i}{\partial z_i^\theta} \right| \right] + \sum_{i=3}^{n-1} \left[ \log p_{Z_\varphi}(z_i^\varphi) + \log \left| \frac{\partial \varphi_i}{\partial z_i^\varphi} \right| \right], \tag{12}
$$

where latent variables $z_i^a, z_i^d, z_i^\theta, z_i^\varphi$ are computed by reversing the mapping in Eqn. (4) and (7), and $p_{Z_a}, p_{Z_d}, p_{Z_\theta}, p_{Z_\varphi}$ are all standard Gaussian distributions. Besides, the atom-wise classifier in the G-SphereNet model, which is used for the focal atom selection, is trained by the binary cross entropy loss. The ground-truth label is 1 if the atom is not valence full filled, otherwise is 0. We describe the detailed training and generation algorithm of G-SphereNet in Appendix B.

## 4 EXPERIMENTS

In this section, we evaluate the proposed G-SphereNet method on the random molecular geometry generation task and the targeted molecule discovery task described in Sec. 2.3. We show that in these tasks, G-SphereNet can outperform previous 3D molecular geometry methods, including G-SchNet (Gebauer et al., 2019) and E-NFs (Satorras et al., 2021a). Note that we do not compare with recent methods (Xu et al., 2021b;a; Shi et al., 2021) of generating 3D molecular geometries from 2D information because they cannot do targeted molecule discovery (Sec. 2.2). In addition, we conduct extensive ablation studies to evaluate the advantages of some designs in G-SphereNet method.

### 4.1 RANDOM MOLECULAR GEOMETRY GENERATION

**Data.** For the random molecular geometry generation task, we evaluate G-SphereNet on the QM9 (Ramakrishnan et al., 2014) dataset. The QM9 dataset provides over 130k molecules and their corresponding 3D molecular geometries computed by density functional theory (DFT). We randomly select 100k 3D molecular geometries as the training data and 10k 3D molecular geometries as the validation data. For fair comparison, the models of our G-SphereNet method and all other methods are trained with the same data split.

**Setup.** We use the chemical validity percentage (**Validity** in short) to evaluate the generation accuracy of G-SphereNet. Specifically, all generated 3D molecular geometries are converted to molecular graphs by the method proposed in Kim & Kim (2015), and the Validity is defined as the percentage of molecular graphs which do not violate chemical valency rules. In addition to the chemical validity, we also evaluate the 3D structural accuracy of the generated molecular geometries. We ever tried to follow G-SchNet (Gebauer et al., 2019) to compute the aligned coordinate differences between the generated geometries and their relaxed ones, but we find the relaxing process involves the expensive computation based on DFT, which takes hours for a single molecular geometry. Hence, we propose to evaluate by the Maximum Mean Discrepancy (**MMD**) (Gretton et al., 2012) distances of bond length distributions. Formally, for a certain type of bond, we obtain its length distribution in the generated geometries and in the geometries of the dataset, separately, and compute the statistical discrepancy between the two length distributions with the MMD distance. We compute the MMD

Table 2: Comparison of different methods on the targeted molecule discovery task. Here ↓ means that our objective is minimizing the property score, while ↑ is maximizing the property score.

| Method | HOMO-LUMO gap↓ | | | Isotropic polarizability↑ | | |
|---|---|---|---|---|---|---|
| | Mean | Optimal | Good percentage | Mean | Optimal | Good percentage |
| QM9 (Dataset) | 6.833 | 0.669 | 3.20% | 75.19 | 196.62 | 2.04% |
| G-SchNet | 3.447 | 0.583 | 78.45% | 76.94 | 204.04 | 30.18% |
| G-SphereNet | **2.907** | **0.294** | **82.73%** | **88.18** | **349.98** | **35.49%** |

distances on six most frequently appeared types of chemical bonds, including carbon-carbon single bonds (**C-C**), carbon-nitrogen single bonds (**C-N**), carbon-oxygen single bonds (**C-O**), hydrogen-carbon single bonds (**H-C**), hydrogen-nitrogen single bonds (**H-N**), and hydrogen-oxygen single bonds (**H-O**). All metrics are computed from 10,000 generated molecular geometries.

The implementation of the SphereNet (Liu et al., 2022) model used for condition information extraction is based on the code of DIG (Liu et al., 2021b) package. We use Adam (Kingma & Ba, 2015) optimizer to train the G-SphereNet model for 100 epochs, with a batch size of 64 and a learning rate of 0.0001. See Appendix C for model configuration and other training details. G-SphereNet is compared with G-SchNet (Gebauer et al., 2019) and E-NFs (Satorras et al., 2021a) in terms of Validity and MMD distances, and we run the code provided by authors to obtain the results of two baseline methods. We do not compare with EDMNet (Hoffmann & Noé, 2019) or 3DMolNet (Nesterov et al., 2020) because the authors of 3DMolNet do not provide their implementation, and the EDMNet model cannot be trained on molecular geometries with variable numbers of atoms, which inhibits a fair comparison with G-SphereNet.

**Results.** We present the performance of different methods in Table 1. Our G-SphereNet method achieves the highest Validity of 88.18%, while E-NFs achieves a much lower Validity of 39.77%. The good performance strongly demonstrates that the sequential fashion of G-SphereNet helps the model to capture the dependency between atoms and learn the underlying chemical rules of molecular geometries more effectively. In addition, compared with G-SchNet, our G-SphereNet achieves lower MMD distances for 4 types of chemical bonds, which shows that our method can model the 3D structural distribution of molecular geometries more accurately. We visualize some molecular geometries generated by G-SphereNet in Figure 2 of Appendix D.

## 4.2 TARGETED MOLECULE DISCOVERY

**Setup.** In the targeted molecule discovery task, we aim to maximize or minimize the expected quantum property score. We conduct two targeted molecule discovery experiments, namely minimizing the HOMO-LUMO gap and maximize the isotropic polarizability. Following G-SchNet (Gebauer et al., 2019), we fine-tune the G-SphereNet model that has been trained in Sec. 4.1 on the biased datasets. Specifically, from the QM9 dataset, we collect all molecular geometries whose HOMO-LUMO gaps are lower than 4.5 eV and all molecular geometries whose isotropic polarizabilities are higher than 91 Bohr$^3$. The G-SphereNet model is then fine-tuned on these two biased datasets so as to generate molecular geometries with low HOMO-LUMO gaps or high isotropic polarizabilities, respectively. Details about the model fine-tuning process are summarized in Appendix C.

In this task, we evaluate the performance by three statistic metrics over the quantum property scores of generated molecular geometries. Specifically, we generate 1000 molecular geometries with the trained model, and filter out the geometries that are not chemically valid. Afterwards, the PySCF (Sun et al., 2018; 2020) package is used to compute the quantum property scores of valid molecular geometries. The performance is then evaluated by three statistic metrics over these quantum property scores. Formally, we calculate the **mean** and the **optimal** value over all property scores, and the percentage of property scores falling into the good region (**good percentage** in short). The good region is defined as scores lower than 4.5 eV for the HOMO-LUMO gap and scores higher than 91 Bohr$^3$ for the isotropic polarizability, respectively. We find that E-NFs (Satorras et al., 2021a) fails to generate enough chemically valid molecular geometries and produce reliable results after fine-tuning, so we only compare our G-SphereNet method with G-SchNet in this task.

Table 3: Results of ablation studies evaluated by Validity. (a) Comparison between using local or/and global features; (b) Comparison of using different 3D information; (c) Comparison of the focal atom selection by the Sigmoid method (ours) and the Softmax method (Simm et al., 2020).

| (a) | | | | (b) | | | | | (c) | |
|---|---|---|---|---|---|---|---|---|---|---|
| Local | Global | Validity | | Distance | Angle | Torsion | Validity | | Setting | Validity |
| ✓ | | 79.93% | | ✓ | | | 74.20% | | Softmax | 67.71% |
| | ✓ | 76.39% | | ✓ | ✓ | | 82.12% | | Sigmoid | **88.18%** |
| ✓ | ✓ | **88.18%** | | ✓ | ✓ | ✓ | **88.18%** | | | |

**Results.** Results of targeted molecule discovery for two quantum properties are summarized in Table 2. For both properties, our G-SphereNet can outperform G-SchNet in all metrics, showing that G-SphereNet can generate more molecular geometries with good properties. Since the two methods use the same pretraining and fine-tuning pipeline, we argue that the better performance of G-SphereNet indicates its stronger ability to search molecular geometries with desirable properties. We illustrate some generated molecular geometries with good properties in Figure 3 of Appendix D.

## 4.3 ABLATION STUDIES

In previous sections, we have demonstrated the effectiveness of our G-SphereNet method on two 3D molecular geometry generation tasks. However, it is unclear whether some designs in our method, such as the use of global features extracted by multi-head attention networks, can indeed lead to good performance. Hence, we conduct extensive ablation studies justifying the use of both local and global features, and the consideration of distance, angle, and torsion information in the 3D conditional information extraction of G-SphereNet. In addition, we study the effects of different focal atom selection methods. In each ablation study, different variants of G-SphereNet models are trained with the setting in Sec. 4.1 and evaluated by the Validity metric. Table 3 show all the results of ablation studies.

**Ablation on local and global feature.** We compare with G-SphereNet variants which only use local features, i.e., node embeddings extracted by SphereNet, and only use global features, i.e., outputs from multi-head attention networks) in Eqn. (10) and (11). Results in Table 3(a) show that using only local or global features both achieves a worse performance.

**Ablation on 3D information.** To show the advantages of using comprehensive 3D information, we replace the SphereNet by SchNet (Schütt et al., 2017) considering only distance information, and DimeNet++ (Klicpera et al., 2020) considering only distances and angles, respectively. As presented in Table 3(b), missing partial 3D information leads to performance degradation.

**Ablation on focal atom selection.** In G-SphereNet, an atom-wise MLP and sigmoid function performs the atom-wise binary classification finding atoms that are not valence full filled. Then the focal atom is randomly selected from those atoms. In contrast, Simm et al. (2020) proposes to directly select the focal atom by an MLP and softmax function. We discuss the differences between two methods in detail in Appendix C. To demonstrate the benefits of our method, we compare it with a G-SphereNet variant replacing our focal atom selection method by the one in Simm et al. (2020). We denote our method by Sigmoid and the method in Simm et al. (2020) by Softmax. As shown in Table 3(c), our Sigmoid method can achieve much better performance than the Softmax method.

## 5 CONCLUSION

We propose G-SphereNet, a novel autoregressive flow model for 3D molecular geometry generation from scratch. G-SphereNet employs a sequential generation pipeline, in which the 3D positions of atoms are obtained through generating the relative distances, angles, and torsion angles. It is flexible and efficient, and can ensure both equivariance and invariance properties. In addition, spherical message passing and attention mechanism are used to extract conditional information during sequential generation. We empirically demonstrate that, compared with previous methods, our G-SphereNet method models the distribution of 3D molecular geometries more accurately, and has stronger capacity to search molecules with good properties. In the future, we will apply our G-SphereNet to more complicated 3D structures, such as proteins and many-body particle systems.

## REPRODUCIBILITY STATEMENT

We have included the source codes of our method in DIG (Liu et al., 2021b) library and provided experiment details in Appendix C for reproducing the results.

## ACKNOWLEDGMENTS

We thank Limei Wang for her help on explaining implementation details of the SphereNet model and Xuan Zhang for his guidance on using PySCF package. This work was supported in part by National Science Foundation grant IIS-1908220.

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

## A PROOF OF THE EQUIVARIANCE AND INVARIANCE

In G-SphereNet, at the $i$-th generation step, we generate the relative distance $d_i$, $\theta_i$, and $\varphi_i$ by the model as

$$d_i, \theta_i, \varphi_i = g\left(z_i^d, z_i^\theta, z_i^\varphi; A_i, R_i\right). \tag{13}$$

Without loss of generality, here we assume $i \geq 3$. Note that $g$ is symmetry-invariant because we extract features using symmetry-invariant SphereNet (Liu et al., 2022) model, hence we have

$$d_i, \theta_i, \varphi_i = g\left(z_i^d, z_i^\theta, z_i^\varphi; A_i, R_iQ^T + \mathbf{1}b^T\right) \tag{14}$$

for any orthogonal matrix $Q \in \mathbb{R}^{3\times3}$ and translation vector $b \in \mathbb{R}^3$. The coordinate $r_i$ of the new atom is then computed from $d_i, \theta_i, \varphi_i$ and the coordinates $r_f, r_c, r_e$. Next, we show that this generation process satisfies the equivariance and invariance properties.

**Theorem 1.** *If $r_i = g^r(z_i^r; A_i, R_i)$, then we have $Qr_i + b = g^r\left(z_i^r; A_i, R_iQ^T + \mathbf{1}b^T\right)$ for any orthogonal matrix $Q \in \mathbb{R}^{3\times3}$ and translation vector $b \in \mathbb{R}^3$.*

*Proof.* Here, we can get that $z_i^r = (z_i^d, z_i^\theta, z_i^\varphi)$, and

$$g^r(z_i^r; A_i, R_i) = h\left(g\left(z_i^d, z_i^\theta, z_i^\varphi; A_i, R_i\right); r_f, r_c, r_e\right),$$

where the function $h(d, \theta, \varphi; r_f, r_c, r_e)$ is defined as

$$h(d, \theta, \varphi; r_f, r_c, r_e) = r_f + \frac{d\cos\theta(r_c - r_f)}{\|r_c - r_f\|_2^2} + \frac{d\sin\theta(r_{e,\varphi} - r_{e,cf})}{\|r_{e,\varphi} - r_{e,cf}\|_2^2}.$$

In this equation, $r_{e,cf}$ is the coordinate of the projection of $e$ on the line $(r_f, r_c)$, and $r_{e,\varphi}$ is the coordinate of $e$ after rotating the plane $(r_f, r_c, r_e)$ along the line $(r_f, r_c)$ by the torsion angle $\varphi$.

For any orthogonal matrix $Q \in \mathbb{R}^{3\times3}$, because $\|Qr\|_2^2 = r^TQ^TQr = r^Tr = \|r\|_2^2$, we have

$$
\begin{aligned}
&h(d, \theta, \varphi; Qr_f + b, Qr_c + b, Qr_e + b) \\
=&Qr_f + b + \frac{d\cos\theta(Qr_c - Qr_f)}{\|Qr_c - Qr_f\|_2^2} + \frac{d\sin\theta(Qr_{e,\varphi} - Qr_{e,cf})}{\|Qr_{e,\varphi} - Qr_{e,cf}\|_2^2} \\
=&Qr_f + b + Q\frac{d\cos\theta(r_c - r_f)}{\|r_c - r_f\|_2^2} + Q\frac{d\sin\theta(r_{e,\varphi} - r_{e,cf})}{\|r_{e,\varphi} - r_{e,cf}\|_2^2} \\
=&Q\left[r_f + \frac{d\cos\theta(r_c - r_f)}{\|r_c - r_f\|_2^2} + \frac{d\sin\theta(r_{e,\varphi} - r_{e,cf})}{\|r_{e,\varphi} - r_{e,cf}\|_2^2}\right] + b \\
=&Qh(d, \theta, \varphi; r_f, r_c, r_e) + b.
\end{aligned}
\tag{15}
$$

Combining the conclusions from both Eqn. (14) and (15), we can easily get that $Qr_i + b = g^r\left(z_i^r; A_i, R_iQ^T + \mathbf{1}b^T\right)$ holds for any orthogonal matrix $Q \in \mathbb{R}^{3\times3}$ and translation vector $b \in \mathbb{R}^3$. $\qquad\square$

**Theorem 2.** *For any orthogonal matrix $Q \in \mathbb{R}^{3\times3}$ and translation vector $b \in \mathbb{R}^3$, we have $p\left(Qr_i + b | A_i, R_iQ^T + \mathbf{1}b^T\right) = p\left(r_i | A_i, R_i\right)$.*

*Proof.* We can easily find that the relative distance $d_i$, angle $\theta_i$, and torsion angle $\varphi_i$ will not change if we transform $r_i$ and $R_i$ with the same $Q$ and $b$. In addition, from Eqn. (14), we have

$$z_i^d, z_i^\theta, z_i^\varphi = g^{-1}(d_i, \theta_i, \varphi_i; A_i, R_iQ^T + \mathbf{1}b^T). \tag{16}$$

Hence, the corresponding latent variables $z_i^d, z_i^\theta, z_i^\varphi$ are invariant to the rotation and translation transformation. Since $p(r_i | A_i, R_i) = p_{Z_d}(z_i^d | A_i, R_i)p_{Z_\theta}(z_i^\theta | A_i, R_i)p_{Z_\varphi}(z_i^\varphi | A_i, R_i)$, we can get that $p\left(Qr_i + b | A_i, R_iQ^T + \mathbf{1}b^T\right) = p\left(r_i | A_i, R_i\right)$ is right for any orthogonal matrix $Q \in \mathbb{R}^{3\times3}$ and translation vector $b \in \mathbb{R}^3$. $\qquad\square$

## B  GENERATION AND TRAINING ALGORITHM

---

**Algorithm 1** Generation Algorithm of G-SphereNet

---

1: **Input:** G-SphereNet model, latent distribution $p_{Z_a}, p_{Z_d}, p_{Z_\theta}, p_{Z_\varphi}$, maximum number of atoms $n$
2:
3: Initialize molecular geometry $G_1$ with one carbon atom, whose coordinate is $r_0 = [0, 0, 0]$
4: **for** $i = 1$ **to** $n - 1$ **do**
5:     $z_i^a \sim p_{Z_a}$
6:     Generate $a_i$ from $z_i^a$
7:     Get the candidate focal atom set by the atom-wise classifer.
8:     **if** the candidate focal atom set is empty **then**
9:         Output $G_i$
10:     **else**
11:         Random select the focal atom $f$ from the candidate focal atom set
12:     **end if**
13:     $z_i^d \sim p_{Z_d}$, $z_i^\theta \sim p_{Z_\theta}$ (if $i \geq 2$), $z_i^\varphi \sim p_{Z_\varphi}$ (if $i \geq 3$)
14:     Generate $d_i$, $\theta_i$ (if $i \geq 2$), $\varphi_i$ (if $i \geq 3$) from $z_i^d, z_i^\theta, z_i^\varphi$
15:     Get $r_i$ from $d_i, \theta_i, \varphi_i$ and $r_f$.
16:     Add a new node with type $a_i$ and coordinate $r_i$ to $G_i$ and set the updated geometry as $G_{i+1}$
17: **end for**
18: Output $G_n$

---

**Algorithm 2** Training Algorithm of G-SphereNet

---

1: **Input:** Molecular geometry dataset $\mathcal{M}$, G-SphereNet model with trainable parameter $\omega$, latent distribution $p_{Z_a}, p_{Z_d}, p_{Z_\theta}, p_{Z_\varphi}$, learning rate $\alpha$, batch size $B$
2:
3: **repeat**
4:     Sample a batch of $B$ molecular graphs $\mathcal{G}$ from $\mathcal{M}$
5:     $L = 0$
6:     **for** $G \in \mathcal{G}$ **do**
7:         Set $n$ as the number of atoms in $G$
8:         Order the atoms in $G$
9:         **for** $i = 1$ **to** $n - 1$ **do**
10:             Get $a_i, d_i, \theta_i$ (if $i \geq 2$), $\varphi_i$ (if $i \geq 3$)
11:             Get $z_i^a, z_i^d, z_i^\theta$ (if $i \geq 2$), $z_i^\varphi$ (if $i \geq 3$)
12:             $L = L - \log p_{Z_a}(z_i^a) - \log p_{Z_d}(z_i^d)$
13:             $L = L - \log p_{Z_\theta}(z_i^\theta)$ if $i \geq 2$
14:             $L = L - \log p_{Z_\varphi}(z_i^\varphi)$ if $i \geq 3$
15:             Add the binary cross entropy loss for the focal atom selection to $L$
16:         **end for**
17:     **end for**
18:     $L = \frac{L}{B}$
19:     $\omega = \omega - \alpha \nabla_\omega L$
20: **until** $\omega$ is converged
21: Output G-SphereNet model with parameter $\omega$

---

## C  EXPERIMENT DETAILS

**Model configuration.** For conditional information extraction, the input 3D molecular geometry is first processed into a cutoff graph. Specifically, any two nodes whose distances are lower than 5.0 Å are connected. The node features are initialized to the one-hot vectors of atom types and the edge features are initialized by spherical basis functions as in Liu et al. (2022). We use the SphereNet (Liu et al., 2022) model with 4 layers to extract features from the input geometry, where the input embedding size is set to 64 and output embedding size is set to 256. Afterwards, global features are extracted by the multi-head attention network with 4 attention heads. In addition, we employ 6 flow layers. Such model configuration is used for all experiments.

**Training and generation details.** In the random molecular geometry generation task, the G-SphereNet model is trained with Adam optimizer for 100 epochs, where the learning rate is 0.0001 and the batch size is 64. We report the results corresponding to the epoch with the best validation loss. In the target molecule discovery task, the model is fine-tuned with a learning rate of 0.0001, a batch size of 32. The number of training epochs is 40 for the HOMO-LUMO gap and 80 for the isotropic polarizability. During generation, we use temperature parameters in the prior Gaussian distributions. Specifically, we change the standard deviation of the Gaussian distribution to the temperature parameter. We use 0.5 for sampling $z_i^a (i \geq 1)$, 0.3 for sampling $z_i^d (i \geq 1)$, 0.4 for sampling $z_i^\theta (i \geq 2)$, and 1.0 for sampling $z_i^\varphi (i \geq 3)$.

**Focal node selection.** In G-SphereNet, the atom-wise MLP takes the feature of each atom as input, and outputs a binary classification score. We consider the atoms whose scores are higher than 0.5 as being classified to candidate focal atoms, and select the exact focal atom from candidate focal atoms. In our experiments, we do not observe any unstable training for this classifier. This atom-wise classifier is trained to classify whether an atom is not valence full filled so that it can be a candidate for next focal atom selection or not. We believe it is not a hard binary classification task so there is no unstable training around the threshold and the training is not sensitive to the threshold choice. Differently, in Simm et al. (2020), the MLP and softmax function directly outputs the probability of being selected to the focal atom for each atom. This MLP is trained by using the sampled focal atom as targets. From the results in Table 3(c), we can clearly find that our method results in much higher chemical validity than the method in Simm et al. (2020).

# D  MORE EXPERIMENT RESULTS

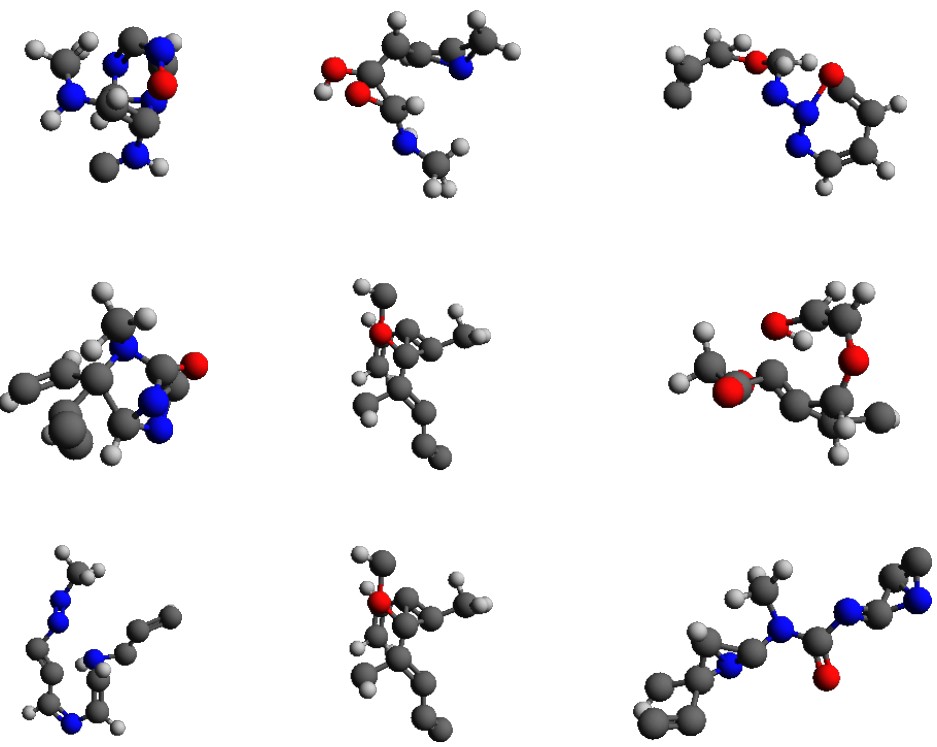

Figure 2: An illustration of the sample molecular geometries generated by the G-SphereNet model trained in the random molecular geometry generation task.

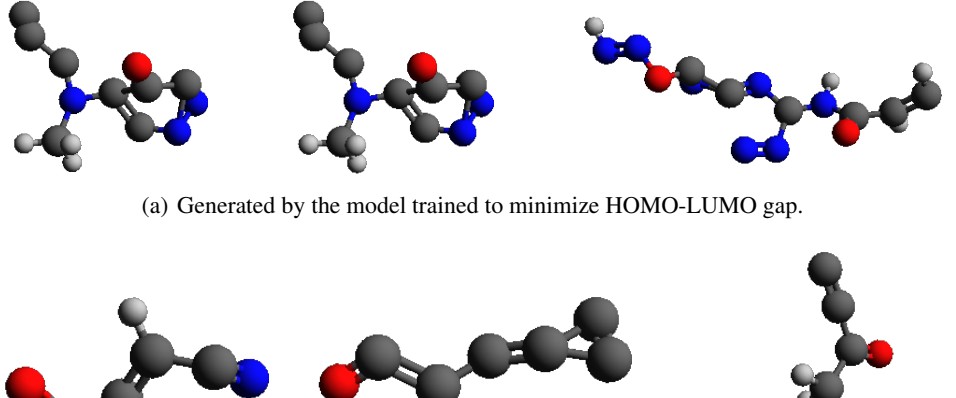

(a) Generated by the model trained to minimize HOMO-LUMO gap.

(b) Generated by the model trained to maximize isotropic polarizability.

Figure 3: An illustration of the sample molecular geometries generated by the G-SphereNet model trained in the targeted molecule discovery task.

