# OpenReview forum: "An Autoregressive Flow Model for 3D Molecular Geometry Generation from Scratch"
_ICLR.cc/2022/Conference — ICLR 2022 Poster_

### Official Review · Reviewer_UA8J · 2021-11-01

**Correctness:** 3
**Technical Novelty And Significance:** 2
**Empirical Novelty And Significance:** 2
**Recommendation:** 5
**Confidence:** 3

**Main Review:**

In G-SphereNet, the authors propose to generate the new atoms using a triplet of (d, theta, phi), which can be seen as spherical coordinates in a local system defined by the focal atom. Such a design seems to simplify the generating process. But such representation has a known issue of Gimbal lock, i.e. when theta = 0, phi no longer affects the atom generation, which I assume happens a lot in the molecule generation process. Despite pairwise distances used in EDMNet/3DMolNet might now indicate a valid 3D point, it doesn't necessarily matter to the generating process as long as (1) there always be a deterministic solution; (2) the distances are SE(3) equivariant/invariant. Compared to G-SchNeet, it makes sense to only generate new atoms near the focal atom. The constraint might serve as a regularizer to generate consistent/meaningful molecules and better capture the dependency between atoms. Most of the benefits listed in section 3.3 remain unproven. In my opinion, regarding the representation of the relative atom position, the author should devise a way to validate the necessity and the benefits of this contribution.

One thing I am confused about is that in Table 3(b), the performance with SchNet backbone is lower than G-SchNet. Does it indicate that the other contributions are actually lowering the performance of G-SchNet?

The organization of the paper structure could be improved. For example, (1) shorten the paragraph that describes the issue of SE(3) invariant in the traditional autoregression flow model; (2) there are two paragraphs describing the focal atom selection (Sec 3.2 and Sec 3.4), which brings confusion about whether the two steps use different focal atoms.

I am not sure if MMD can substitute the original RMSD metric. As the RMSD between the generated model and the relaxed model evaluates the accuracy of the generated model, while MMD between the generated model and the dataset ignores the molecule structures.

There are a few details not clear to me in the paper:
(1) Sec 2.3, G-SchNet also uses a flexible sequential generation pipeline based on autoregressive flow models?
(2) Sec 3.3, what if c and e are mutually closest points?

Some typos:
Gaussian distribution N(0, I) --> N(0, 1)
Sec 3.5 last sentence: genration --> generation

**Summary Of The Paper:**

The paper addresses the problem of generating 3D molecular geometries.
The proposed G-SphereNet takes SphereNet as the backbone to generate 3D molecular geometries from scratch.
The generation is done in a step-by-step fashion, with each step generating a (distance, angle, torsion) triplet, conditioning on features extracted by an attention mechanism.

**Summary Of The Review:**

Overall, the paper builds largely based on the setting of G-SchNet. There are a few designs to improve the performance. However, the benefits of the designs are not thoroughly/fairly evaluated.

---

> ### Author Response · Authors · 2021-11-11
> **Response to Reviewer UA8J Part 1**
>
> Thank you for your constructive feedback and comments. We believe your concerns and questions can be addressed by our responses below. We hope you can increase your score if we address your concerns.
>
> **Q1**: *The $(d, \theta, \phi)$ representation has a known issue of Gimbal lock, i.e. when $\theta = 0$, $\phi$ no longer affects the atom generation, which I assume happens a lot in the molecule generation process.*
>
> **Response**: Thank you for this insightful opinion! We agree that theoretically, Gimbal lock exists for some extreme cases, such as $\theta=0$. However, we find such Gimbal lock phenomena do not happen during generation. We think that is because the Gimbal lock phenomena do not exist in the molecular geometry data fed to the model during training, then a well trained model is supposed to have learned to generate valid 3D representations. In other words, the probability of Gimbal lock will become very low after the model is trained from the data without Gimbal lock. Hence, we believe Gimbal lock is not a significant issue for our method, but it is meaningful to think about designing a generation pipeline that can totally avoid Gimbal lock in our future work.
>
> **Q2**: *Despite pairwise distances used in EDMNet/3DMolNet might now indicate a valid 3D point, it doesn't necessarily matter to the generating process as long as (1) there always be a deterministic solution; (2) the distances are SE(3) equivariant/invariant.*
>
> **Response**: We agree that the distances are SE(3) equivariant/invariant, but we believe obtaining valid 3D coordinates of atoms are necessary to many practical applications, such as computing molecular properties. For instance, to compute the quantum properties of molecules, we need the 3D coordinates of atoms as the input to density functional theory based algorithms. Hence, we argue that being able to generate valid 3D coordinates is significant for 3D molecular geometry generation. Our G-SphereNet method is better solution than EDMNet and 3DMolNet in that it can obtain the exact 3D coordinates of atoms, while EDMNet and 3DMolNet only generate Euclidean distance matrices and are not guaranteed to find corresponding 3D coordinates for atoms.
>
> **Q3**: *Most of the benefits listed in section 3.3 remain unproven. In my opinion, regarding the representation of the relative atom position, the author should devise a way to validate the necessity and the benefits of this contribution.*
>
> **Response**: (1) The benefits of our method over EDMNet and 3DMolNet are that EDMNet and 3DMolNet are not able to generate valid 3D coordinates while our method can always obtain exact 3D coordinates. The ability of producing valid 3D coordinates is necessary (see the above response to Q2) and the limitations of EDMNet and 3DMolNet can be proven theoretically by the theorem in [ref1]. In addition, we believe that the benefits of our method over G-SchNet and E-NFs can be demonstrated by our experimental results in Section 4.1 and 4.2. (2) In our opinion, the necessity and benefits of the relative atom position can be validated conceptually and empirically. Conceptually, as we have extensively discussed in Section 3.2, directly generating the absolute atom position with autoregressive flow models does not satisfy the invariance and equivariance properties. To solve this problem, we think it is necessary and beneficial to generate the relative position in the form of distances, angles, and torsion angles. Empirically, we validate the benefits of the relative atom position in the experimental results in Section 4.1. Our method can achieve significantly better performance than the E-NFs method, and the E-NFs method generates the absolute atom position. We believe this can demonstrate the benefits of the relative atom position.
>
> [ref1]: Dokmanic, Ivan, et al. "Euclidean distance matrices: essential theory, algorithms, and applications." IEEE Signal Processing Magazine 32.6 (2015): 12-30.

---

> ### Author Response · Authors · 2021-11-11
> **Response to Reviewer UA8J Part 2**
>
> **Q4**: *One thing I am confused about is that in Table 3(b), the performance with SchNet backbone is lower than G-SchNet. Does it indicate that the other contributions are actually lowering the performance of G-SchNet?*
>
> **Response**: We agree that the performance with SchNet backbone is lower than G-SchNet. Such phenomenon can be intuitively explained that if the input conditional information only contains distances, it is better to only obtain distances rather than obtain complete 3D representations of distances, angles, and torsion angles. But we would like to emphasize that **our method is not an extension or incremental work of G-SchNet**. We do not aim to improve G-SchNet but we aim to develop a completely different 3D molecular geometry generation framework, which has a novel sequential generation pipeline and conditional information extraction method. Results in Table 3(b) are used to demonstrate that extracting information of distances, angles, and torsion angles is the best choice, and in Table 1 we show that our method can achieve better performance than G-SchNet with all novel contributions.
>
> **Q5**: *The organization of the paper structure could be improved. For example, (1) shorten the paragraph that describes the issue of SE(3) invariant in the traditional autoregression flow model; (2) there are two paragraphs describing the focal atom selection (Sec 3.2 and Sec 3.4), which brings confusion about whether the two steps use different focal atoms.*
>
> **Response**: Thanks for your suggestion. (1) We think this part is a significant motivation to our proposed method of generating 3D position of new atoms, so we describe it in detail and try to present our motivation clearly. In addition, we believe it is easy for readers not familiar with this topic to follow and understand if we make a comprehensive description of the SE(3) invariant issue. Hence, we would like to keep the original paragraph. (2) We agree that confusion may exist. The two paragraphs in Section 3.2 and Section 3.4 are describing the same focal atom selection process. To avoid confusion, we remove the sentence in Section 3.2 and only describe the details of focal atom selection in the second to last paragraph of Section 3.4 in the revision.
>
> **Q6**: *I am not sure if MMD can substitute the original RMSD metric. As the RMSD between the generated model and the relaxed model evaluates the accuracy of the generated model, while MMD between the generated model and the dataset ignores the molecule structures.*
>
> **Response**: We agree that MMD cannot substitute the original RMSD metric. The reason why we use MMD is the limitation of our computation resources, which is clearly presented in the second paragraph of Section 4.1 — “We ever tried to follow G-SchNet (Gebauer et al., 2019) to compute the aligned coordinate differences between the generated geometries and their relaxed ones, but we find the relaxing process involves the expensive computation based on DFT, which takes hours for a single molecular geometry. Hence, we propose to evaluate by the Maximum Mean Discrepancy (MMD) (Gretton et al., 2012) distances of bond length distributions.” It is true that MMD is not the best metric to evaluate the accuracy of 3D structures, but it can evaluate how similar the bond lengths of the generated 3D molecular geometries are to those of real molecules. We hope you can understand our difficulties and fairly evaluate our experiments.

---

> ### Author Response · Authors · 2021-11-11
> **Response to Reviewer UA8J Part 3**
>
> **Q7**: *There are a few details not clear to me in the paper: (1) Sec 2.3, G-SchNet also uses a flexible sequential generation pipeline based on autoregressive flow models? (2) Sec 3.3, what if c and e are mutually closest points?*
>
> **Response**: (1) No, G-SchNet uses a sequential generation pipeline but does not use autoregressive flow models. (2) There is not any problem if $c$ and $e$ are mutually closest points.  As the paper described in Section 3.2 (the last 3 lines of page 4), $c$ is the atom closest to $f$, meaning that $c=\arg\min_{0\le j < i, j\ne f} ||r_j - r_f||_2$. It is totally OK if $c$ and $e$ are mutually closest point and $c$ is the closest point to $f$ simultaneously, in which case $||r_c - r_e||_2 \le ||r_c -r_f ||_2$ and $||r_c - r_f||_2 \le ||r_j - r_f||_2, 0\le j < i, j\ne f, j\ne c$. Note that because $e$ is selected from the atoms different from $f$ (we add this description in the revision), our method still works even if $c$ and $f$ are mutually closest points.
>
> **Q8**: *Some typos: Gaussian distribution N(0, I) --> N(0, 1) Sec 3.5 last sentence: genration --> generation*
>
> **Response**: Thank you very much for pointing out typos! We have corrected them in the revision.
>
> **Q9**: *Overall, the paper builds largely based on the setting of G-SchNet. There are a few designs to improve the performance. However, the benefits of the designs are not thoroughly/fairly evaluated.*
>
> **Response**: (1) We do not agree that our method builds largely based on the setting of G-SchNet. In fact, other than that both methods generate new atoms sequentially one at a time, all other settings are different for the two methods. First, in G-SchNet, the position of a new atom is selected from a limited candidate set of grid points, in which the probability for each grid point is computed through predicting the probability of discretized distances by models. However, our method determines the positions of new atoms through generating distances, angles, and torsion angles through autoregressive flow models. Second, G-SchNet only extracts distance features for conditional information. But our method extracts features of distances, angles, and torsion angles, and we add global features extracted by multi-head attention networks. In a word, the generation and conditional information extraction part of our method are totally different from G-SchNet. (2) We have shown comprehensive experimental results in Section 4. We show that our method can achieve better performance than previous 3D geometry generation methods, G-SchNet and E-NFs, in Section 4.1 and 4.2. And we study the benefits of multiple novel designs in our method through ablation studies in Section 4.3. We agree that our current experiments are not 100% complete and thorough, but they are sufficient to demonstrate the benefits of our method.

---

> ### Author Response · Authors · 2021-11-19
> **Looking forward to your feedback**
>
> Dear Reviewer UA8J,
>
> Thank you again for your valuable comments and suggestions. We have posted our response to your initial comments. We are looking forward to your feedback and willing to answer your further questions.
>
> Thank you!
>
> Authors

---

> ### Author Response · Authors · 2021-11-28
> **Could you kindly check our response and revision?**
>
> Dear Reviewer UA8J,
>
> Since the discussion period will end soon, could you kindly check our response and revision? Currently, other three reviewers are all positive about our work. We are looking forward to knowing if we have properly addressed your concerns.
>
> Thank you!
>
> Authors

---

### Official Review · Reviewer_euwK · 2021-11-02

**Correctness:** 3
**Technical Novelty And Significance:** 3
**Empirical Novelty And Significance:** 2
**Recommendation:** 6
**Confidence:** 3

**Main Review:**


Overall the paper does a good job of extending autoregressive flow models to support full 3D molecular geometry. The extension depends on a few relevant insights in how to best parameterize the problem and shows good overall results and improvements over prior methods. As this is not a commonly explored area and outside my expertise, it is hard to judge how significant the improvmeents in Table 2 and related results are, but from my limited understanding they seem sufficiently significant.

"These conditional generation methods assume that we are given the target molecules in the form of molecular graphs, but we do not know their 3D geometries. However, this assumption does not hold in the targeted molecule discovery problem." -- it seemed to me when I read this that it should still be possible to still do targeted molecule discovery, by first generating the molecular graph then generate 3D geometry from the graph  and applying the loss to the generated 3D geometry?

Several times the option of "is randomly chosen from the atoms whose classification scores are higher than 0.5" is taken, but it's not really described how this operates when there are no classification scores higher than 0.5; and furthermore it seems like this type of classifier might make network training unstable around the threshold, so it would be good to discuss how sensitive training is to metaparameter choice.

I found the ablations useful to demonstrate the different components, and I think molecular design is potentially an important area to focus on more.


**Summary Of The Paper:**

The paper focuses on a generative model for 3D molecules that generates full 3D molecular gemoetry, instead of generating a molecular graph and then generating the 3D molecular geometry conditional on the graph. The method using an autoregressive flow model, which has been applied to molecular graph generation but not full geometry. This enables some new applications such as molecular design to target a geometry-dependent objective function.


**Summary Of The Review:**


The paper extends autoregressive flow models for 3D molecular generation to also generative 3D geometry. This helps with some challenging molecular discovery tasks, and a set of ablations and comparisons demonsrtate the effectiveness of the method.  The main concern is that it is somewhat incremental over the most similar prior work which does not generate 3D molecular geometry.

---

> ### Author Response · Authors · 2021-11-11
> **Response to Reviewer euwK**
>
> Thank you for agreeing that our work is insightful and achieves good overall results and improvements over previous methods! We believe your concerns and questions can be addressed by the following responses.
>
> **Q1**: *"These conditional generation methods assume that we are given the target molecules in the form of molecular graphs, but we do not know their 3D geometries. However, this assumption does not hold in the targeted molecule discovery problem." – It seemed to me when I read this that it should still be possible to still do targeted molecule discovery, by first generating the molecular graph then generating 3D geometry from the graph and applying the loss to the generated 3D geometry?*
>
> **Response**: In the targeted molecule discovery problem, we focus on quantum properties of molecules. If we search targeted molecules by first generating the molecular graph then generating 3D geometry from the graph, we are exactly searching molecules with good quantum properties in the molecular graph space. However, there is no way to obtain quantum properties from molecular graphs so we have to compute them from the subsequently generated 3D geometries from molecular graphs. The problem is that the process from molecular graphs to 3D geometries here is **not deterministic**, thus we may get multiple very different quantum property values for the same molecular graph. In other words, **we cannot find a one-to-one mapping from molecular graphs to quantum properties**, thereby making the search for molecules with good quantum properties in the molecular graph space impossible. We believe the targeted molecule discovery based on molecular graphs can only be achieved if our target properties can be deterministically obtained from molecular graphs, which is not the case for quantum properties.
>
> **Q2**: *It's not really described how the focal atom selection operates when there are no classification scores higher than 0.5; and furthermore it seems like this type of classifier might make network training unstable around the threshold, so it would be good to discuss how sensitive training is to metaparameter choice.*
>
> **Response**: (1) When there are no classification scores higher than 0.5, the sequential generation process will terminate since no atom can be chosen as the focal atom. We have added this description in the second to last paragraph of Section 3.4 in the revision – “If all the classification scores outputted from the atom-wise classifier are lower than 0.5, then no atom can be chosen as the focal atom and the sequential generation process terminates.”  (2) In our experiments, we do not observe any unstable training for this classifier. This atom-wise classifier is trained to classify whether an atom is not valence full filled so that it can be a candidate for next focal atom selection or not. We believe it is not a hard binary classification task so there is no unstable training around the threshold and the training is not sensitive to the threshold choice. We have added the above discussion of the threshold in Appendix C of the revision.
>
> **Q3**: *The main concern is that it is somewhat incremental over the most similar prior work which does not generate 3D molecular geometry.*
>
> **Response**: It is **not trivial** to extend the prior molecule generation work based on autoregressive flow models to the generation of much more complicated 3D molecular geometries. (1) As we extensively discussed in Section 3.2, the generation of 3D positions of atoms have to consider the invariance and equivariance properties, which is not the case if only atom types and bonds are needed to be generated. We propose an effective and flexible sequential generation scheme that generates distances, angles, and torsion angles to obtain the 3D positions of atoms. It can not only naturally fit into the autoregressive flow model, but also theoretically ensure the invariance and equivariance properties. (2) We propose to use expressive SphereNet and multi-head attention network models to effectively extract conditional information from the previously generated 3D geometries. We believe these are very significant novelty contributions so our method is not a simple incremental work over the prior work which does not generate 3D molecular geometries.

---

> ### Author Response · Authors · 2021-11-19
> **Looking forward to your feedback**
>
> Dear Reviewer euwK,
>
> Thank you again for your valuable comments and suggestions. We have posted our response to your initial comments. We are looking forward to your feedback and willing to answer your further questions.
>
> Thank you!
>
> Authors

---

> ### Comment · Reviewer_euwK · 2021-11-28
>
> Thanks for your updates here.  Having read the reviews and responses, I remain in favor of accepting the paper, given the changes in response to reviewer comments.  I think advancing generative models over molecular design remains an important problem.

---

### Official Review · Reviewer_JRzq · 2021-11-02

**Correctness:** 3
**Technical Novelty And Significance:** 3
**Empirical Novelty And Significance:** 3
**Recommendation:** 8
**Confidence:** 3

**Main Review:**

The main strength of the paper is in (i) showing that generating a 3D molecular structure from scratch is feasible, and (ii) in demonstrating that a sequential approach is particularly effective for this task. Both aspects are novel to my knowledge, and worthy of communication to the relevant community.

The paper is fluent, clear, and easy to follow. The positioning with respect to the literature is also rather clear, and the analogies and differences with existing approaches are well described. The experimental section is well executed, and includes an extensive ablation study confirming the validity of the proposed pipeline in its entirety. Comparisons with very recent methods addressing the same task are included. Code is also included.

Particularly significant are the quantitative results, which demonstrate an important gap in accuracy compared with other approaches (>10% in most cases).

One aspect that I feel might benefit from additional discussion is on the rationale behind a sequential model being more effective for generation. The paper attempts to give some motivations (e.g. that it helps to capture the relation between atom and density more effectively), but additional insight would be useful. If the authors can share more of their insight during the discussion period, it would help my understanding of the approach.

Perhaps more importantly, something that does not really exude from the paper is the potential impact that the method could have on practical applications. The conclusions mention that a promising future direction would be the generation of more complex molecules (such as proteins), but at the current stage, what is the main benefit of this method for downstream tasks in structural biology? If this is to be taken as a first step toward practical applications is also fine, but it would be nice to see this clarified in the manuscript.

As a minor comment, it is not clear to me how Figures 2 and 3 (in the supplementary material) should be interpreted. Is there a way to read these figures, that suggests that the generative model is doing a good job at synthesizing new molecules?

**Summary Of The Paper:**

The paper introduces a new generative model for 3D molecular geometry, taking the form of a graph embedded in 3D where each node is an atom, and each edge is a bond. Differently from existing methods, the proposed pipeline attempts to generate the 3D molecule from scratch, instead of starting, e.g., from a 2D molecular graph as other methods do. The pipeline is sequential and is based on autoregressive flow models, which have been previously used for the generation of 2D molecular graphs, but not for the present task. The paper convincingly shows that the proposed model works better than competing approaches for the same task.

**Summary Of The Review:**

This is a well written paper, with a novel contribution that is likely to leave an impact in the emerging field of generative models for molecular structures. The main weakness, in my opinion, lies in the complexity of the generated molecules, which at this point is not clear whether it can lead to relevant practical applications or not. But overall, I am still leaning positive in the light of the gap in accuracy with respect to previous methods.

---

> ### Author Response · Authors · 2021-11-11
> **Response to Reviewer JRzq**
>
> Thank you very much for your positive rating and insightful comments! We hope your questions can be addressed by our responses below.
>
> **Q1**: *Additional insight behind a sequential model being more effective for generation would be useful.*
>
> **Response**: Thank you for this good suggestion! We believe there are two reasons which may explain the effectiveness of sequential models. First, sequential methods explicitly model the dependency relationship between atoms. Concretely, the probability distribution of an individual atom's atomic number and 3D position is somehow influenced by other atoms, and a sequential model can capture such influence by taking the information of existing atoms as input when generating a new atom. However, non-sequential methods, such as [ref1] and [ref2], generate a complete molecule at a time and assume all atoms to be mutually independent, failing to capture the relations between atoms. Second, sequential methods only generate a part of the data at a time, while non-sequential methods generate the complete data at a time. Hence, sequential methods have a much lower dimensional generation space. Because it is easier to capture the density of a lower dimensional space, we believe sequential methods are naturally more effective than non-sequential methods.
>
> [ref1]: Madhawa, Kaushalya, et al. "GraphNVP: An invertible flow model for generating molecular graphs." arXiv preprint arXiv:1905.11600, 2019.
>
> [ref2]: Honda, Shion, et al. "Graph residual flow for molecular graph generation." arXiv preprint arXiv:1909.13521, 2019.
>
> **Q2**: *What is the main benefit and potential impact of this method for downstream tasks in structural biology?*
>
> **Response**: This is a nice question. Our work demonstrates that autoregressive models have the ability to model very complicated 3D molecular geometries, and discover novel 3D molecules with good quantum properties. We believe our method can have potential application value in some practical problems, such as discovering drugs with desired biochemical properties. Currently, our method focuses on small molecules, but we believe it can serve as a good starting point to motivate the application of similar generative methods to more complicated molecules, such as proteins and material molecules.
>
> **Q3**: *It is not clear to me how Figures 2 and 3 (in the supplementary material) should be interpreted. Is there a way to read these figures, that suggests that the generative model is doing a good job at synthesizing new molecules?*
>
> **Response**: The molecular geometries in Figure 2 and 3 are plotted by Avogadro, a software to visualize 3D molecules. Figure 2 and 3 are used to visualize some examples of generated 3D molecular geometries. There is not a specific way to interpret or read these figures to suggest the goodness of molecules.

---

> ### Author Response · Authors · 2021-11-19
> **Looking forward to your feedback**
>
> Dear Reviewer JRzq,
>
> Thank you again for your positive rating and insightful comments! We have posted our response to your initial comments. We are looking forward to your feedback and willing to answer your further questions.
>
> Thank you!
>
> Authors

---

### Official Review · Reviewer_1x5L · 2021-11-03

**Correctness:** 3
**Technical Novelty And Significance:** 3
**Empirical Novelty And Significance:** Not applicable
**Recommendation:** 6
**Confidence:** 3

**Main Review:**

Pro:
1. The paper proposes a good way to handle the invariance issue during generating 3D positions of the atoms in the molecule by outputting related position change (d, theta, phi) instead of outputting absolute coordinate r. They have given thorough reasoning in the paper why they should not use r.

2. The paper has shown good results compared with previous methods and has done several ablation studies to show that different components in their method are useful.

Con:
1. The contribution seems limited. Most of their pipeline is based on previous work including (Liu et al. 2021d) and (Vaswani et al. 2017).

2. The writing of the paper is a little bit messy. Some related algorithm like (Vaswani et al. 2017) is shown with proper math formulations, while some others like the message passing in (Liu et al. 2021d) are missing. It's better to rearrange the paper and make it more self-contained.

**Summary Of The Paper:**

The paper proposes a new way to directly generate 3D molecular geometries from scratch. They output the atoms one by one, including the type of the atom and the related position from one chosen focal atom. They use a SphereNet (Liu et al. 2021d) - a spherical message-passing network and an additional multi-head attention network (Vaswani et al. 2017) to process the existed information in the partial generation results. They have shown in 2 different tasks that they are better than previous methods.

**Summary Of The Review:**

The paper has improvements compared to previous work based on the system design and new algorithms, however, some critical parts seem incremental.

---

> ### Author Response · Authors · 2021-11-11
> **Response to Reviewer 1x5L**
>
> Thank you for positive feedback and valuable comments! We hope your concerns and questions can be addressed by our responses below.
>
> **Q1**: *The contribution seems limited. Most of their pipeline is based on previous work including (Liu et al. 2021d) and (Vaswani et al. 2017).*
>
> **Response**: Our work is fundamentally different from [ref1] and [ref2] in that **while [ref1] and [ref2] developed predictive models, our work is generative**. Though our method uses SphereNet [ref1] and multi-head attention network [ref2] models, they only play the role of extracting conditional information from the geometry generated in the previous step. On the other hand, **the major novelty contribution** of our method is a flexible sequential generation scheme based on flow models, which generates distances, angles, and torsion angles to ensure the equivariance and invariance properties. **It is a much more significant novelty contribution and is totally independent of [ref1] and [ref2]**. In addition, through extensive experiments, we demonstrate our method to be more effective and expressive to model the density of complicated 3D molecular geometries than previous methods. Hence, we believe our contribution is significant in the 3D molecular geometry generation problem, and our method is far more complicated than simply building a generation pipeline upon [ref1] and [ref2].
>
> [ref1]: Liu, Yi, et al. "Spherical message passing for 3d graph networks." arXiv preprint arXiv:2102.05013, 2021.
>
> [ref2]: Vaswani, Ashish, et al. "Attention is all you need." Advances in neural information processing systems, 2017.
>
> **Q2**: *The writing of the paper is a little bit messy. Some related algorithm like (Vaswani et al. 2017) is shown with proper math formulations, while some others like the message passing in (Liu et al. 2021d) are missing. It's better to rearrange the paper and make it more self-contained.*
>
> **Response**: Thank you very much for this insightful suggestion! We have added more detailed descriptions about the SphereNet model in Section 3.4 in the revision. However, since SphereNet is a rather complicated model and its message passing mechanism cannot be simply described by a few sentences, we are not able to provide a complete introduction of SphereNet. We recommend you to read the original paper of SphereNet [ref1] for more details of it.
>
> [ref1]: Liu, Yi, et al. "Spherical message passing for 3d graph networks." arXiv preprint arXiv:2102.05013, 2021.

---

> ### Author Response · Authors · 2021-11-19
> **Looking forward to your feedback**
>
> Dear Reviewer 1x5L,
>
> Thank you again for your positive feedback and comments. We have posted our response to your initial comments. We are looking forward to your feedback and willing to answer your further questions.
>
> Thank you!
>
> Authors

---

### Decision · Program_Chairs · 2022-01-20

**Decision:**

Accept (Poster)

**Comment:**

This work introduces an autoregressive flow model that generates molecular geometries by placing one atom at the time.
In order to preserve the E(3) invariance of the density, successive atom locations are sampled relative to already placed atoms (in a coordinate system described by distance, angle and torsion).
The paper is overall well-written and experimental results are compelling.